

# Temporal predictability does not impact attentional blink performance: effects of fixed vs. random inter-trial intervals

Lucienne Shenfield[1], Vanessa Beanland[2] and Deborah Apthorp[3,4]

[1] Research School of Psychology, Australian National University, Canberra, ACT, Australia
[2] Department of Psychology, University of Otago, Dunedin, New Zealand
[3] School of Psychology, University of New England, Armidale, NSW, Australia
[4] Research School of Computer Science, Australian National University, Canberra, ACT, Australia

## ABSTRACT

**Background**. Does the inclusion of a randomized inter-trial interval (ITI) impact performance on an Attentional Blink (AB) task? The AB phenomenon is often used as a test of transient attention (*Dux & Marois, 2009*); however, it is unclear whether incorporating aspects of sustained attention, by implementing a randomized ITI, would impact task performance. The current research sought to investigate this, by contrasting a standard version of the AB task with a random ITI version to determine whether performance changed, reflecting a change in difficulty, engagement, or motivation.
**Method**. Thirty university students (21 female; age range 18–57, $M_{age} = 21.5$, $SD = 7.4$) completed both versions of the task, in counterbalanced order.
**Results**. No significant difference in performance was found between the standard AB task and the AB task with the random ITI. Bayesian analyses suggested moderate evidence for the null.
**Conclusion**. Temporal unpredictability did not appear to impact task performance. This suggests that the standard AB task has cognitive properties with regards to task difficulty, engagement, and motivation, that are inherently similar to tasks that employ a randomized ITI to measure sustained attention (e.g., the Psychomotor Vigilance Task; PVT; *Dinges & Powell, 1985*). This finding provides important support for future research which may seek to obtain a more detailed understanding of attention through the comparison of performance on transient and sustained attention tasks.

Corresponding author
Deborah Apthorp,
dapthorp@une.edu.au

## INTRODUCTION

Given the important role attention plays in behavioural outcomes (*Carrasco, 2011*; *Correa et al., 2006*), many researchers have sought to better understand attentional processes by examining performance limitations within specific attentional tasks (*Raz, 2004*). This research has encompassed a variety of paradigms and tasks, exploring both spatial (e.g., *Intriligator & Cavanagh, 2001*) and temporal (e.g., *Potter et al., 2013*) characteristics of attention. The present research focuses on the attentional blink (AB), which has been used to investigate temporal aspects of attentional selection (*Dux & Marois, 2009*).

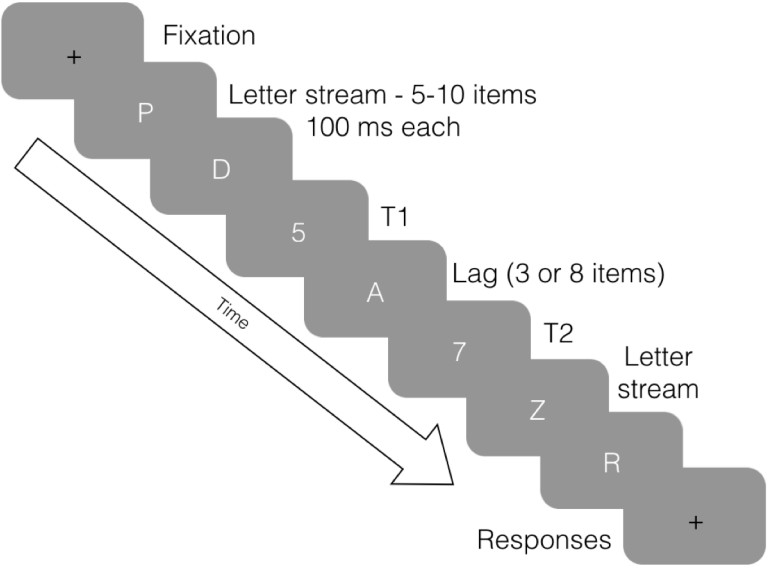

**Figure 1** **Attentional Blink procedure.** Eight frames of an RSVP stream in an attentional blink task. Participants are asked to identify two target stimuli (numbers) within distractor stimuli (letters). The above figure displays T2 presented at a lag of 3 (i.e., three items after T1). In our experiment, the timing was exactly 100 ms for each stimulus (12 frames at 120 Hz), as illustrated here.

The AB occurs when an observer has a reduced ability to perceive a second target (T2) within a set of distractors when it is presented within 800 ms of an initial target (T1; see Fig. 1). The phenomenon was first documented by *Broadbent & Broadbent (1987)*; however, the term 'attentional blink' was coined by *Raymond, Shapiro & Arnell (1992)*. This name does not refer to a physical blink, but rather to the momentary lapse in attentional ability after selection of T1 (*Shapiro, Raymond & Arnell, 1997*). The event is typically measured using a Rapid Serial Visual Presentation (RSVP) task (*Potter & Levy, 1969*). During the task, stimuli are displayed in quick succession, usually around 100 ms apart, in order to test information processing limits. Importantly, *Raymond, Shapiro & Arnell (1992)* noted that detection of T2 significantly improved when subjects were instructed to deliberately ignore T1. This suggests that the AB occurs due to attentional, as opposed to sensory, limitations, validating its use as a measure of attentional performance. The attentional nature of the effect has been further confirmed through numerous AB studies over the past 25 years (see *Dux & Marois, 2009*, for a review).

Past research has investigated many factors thought to influence performance on the AB task, including individual differences (*Colzato et al., 2007*; *Kelly & Dux, 2011*; *MacLean, Arnell & Cote, 2012*; *Martens et al., 2006*). Although individual differences are strongly predictive of AB magnitude, recent research suggests that levels of anticipation can also affect AB performance (*Maclean & Arnell, 2011*). This research has shown that changing temporal characteristics, particularly relating to temporal predictability, can affect blink magnitude. For instance, *Maclean & Arnell (2011)* found that increased attentional blink magnitude was associated with higher levels of anticipatory attention, as measured by
alpha oscillations using electroencephalogram (EEG). This is consistent with the theory that the AB reflects overinvestment in T1 (i.e., excessive anticipation), which leaves the observer unprepared to process T2. Increasing temporal predictability of the AB sequence may therefore attenuate the AB by allowing observers to better prepare for when to allocate attentional resources. There are several aspects of stimulus timing that can be manipulated in an AB task, including foreperiod, target onset asynchrony (TOA) and inter-trial interval (ITI).

## Trial Foreperiod

In the context of AB, foreperiod refers to the time between trial commencement and presentation of T1. The impact of foreperiod on AB was investigated by *Badcock et al. (2013)*, who manipulated the foreperiod by extending, shortening and/or randomising the time between trial onset and T1 presentation. *Badcock et al. (2013)* found that, compared with randomly variable foreperiods, having a predictable foreperiod attenuated the AB, but only for relatively long foreperiods (∼880 ms). They concluded that brief or temporally unpredictable foreperiods contribute to the AB, as observers do not have adequate time to prepare for the dual task of detecting both T1 and T2 (*Badcock et al., 2013*). The consequence is that observers prioritise processing of T1, and require additional time to fully process T2, which results in an AB if T2 follows shortly after T1.

## Target Onset Asynchrony (TOA)

TOA (also called stimulus onset asynchrony or SOA) refers to the length of time in milliseconds between the presentation of T1 and T2. In AB tasks this is commonly referred to as the "lag", which denotes the number of distractors between T1 and T2. Lag is arguably the strongest predictor of T2 perception: T2 is reliably perceived when presented immediately after T1 (i.e., lag 1), making it resistant to the AB effect. This is known as *lag 1 sparing*, a robust feature of the AB (*Hommel & Akyürek, 2005*; *Lunau & Olivers, 2010*; *Martin & Shapiro, 2008*; although see *Visser et al., 1999*). The ability to perceive T2 is typically worst at around lag 3, and steadily improves until around lag 8, when it is usually reliably perceived again (see Fig. 2).

Introducing a consistent lag or TOA (e.g., presenting a block of only lag 3 trials) improves AB task performance, especially when participants are made explicitly aware of the consistency (*Martens & Johnson, 2005*; *Visser, 2015*; *Visser et al., 2014*). Even when lag varies within blocks, providing a lag length cue before the commencement of each trial can reduce blink magnitude (*Martens & Johnson, 2005*). This further demonstrates the important role that anticipation plays in AB performance, with some researchers suggesting that familiarity with these patterns may account for training improvements (*Tang, Badcock & Visser, 2014*), and even learning effects (*Choi et al., 2012*).

## Inter-Trial Interval (ITI)

ITI refers to the time between a participant's response on a trial N−1, and the start of the next trial N. Although AB tasks classically employ a fixed ITI, the present research seeks to explore the role that the ITI plays in shaping task performance. The time between one trial ending and the next trial commencing may be crucial with regard to preparation and
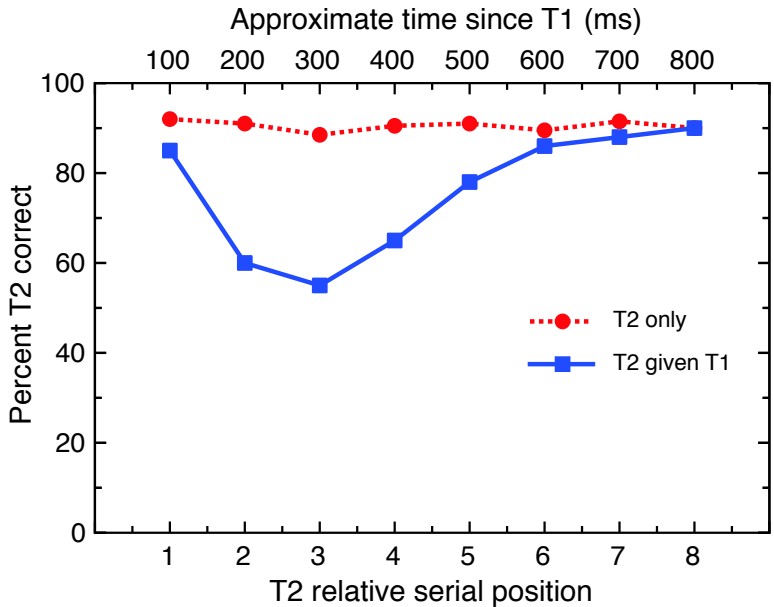

**Figure 2  Typical results for an AB task.** This figure illustrates typical performance on attentional blink tasks, showing lag 1 sparing with intact performance at lag 1 and impaired performance at lags 2–5. Note T2—T1 (blue line) represents correct detection of T2 following correct detection of T1; T2 only (red line) represents correct detection of T2 when the observer is not required to detect T1.

anticipation, and as yet these intervals have not been investigated in relation to the AB. This would be particularly relevant to research investigating the role of pre-stimulus alpha (*Maclean & Arnell, 2011*; *MacLean, Arnell & Cote, 2012*; *Zauner et al., 2012*).

Although historically it has been assumed that ITI does not affect AB performance, ITI has been found to influence both reaction time and accuracy in vigilance tasks, such as the Psychomotor Vigilance Task (PVT; *Dinges & Powell, 1985*) and the Continuous Performance Task (CPT; *Cornblatt et al., 1988*). This raises the question of whether ITI can also affect performance on the AB, a temporal attention task which usually requires unspeeded responses.

## The Current Study

Previous studies investigating the temporal predictability of the AB have manipulated temporal attributes within the RSVP stream. Usually this is achieved by identifying a temporal aspect which is typically variable (e.g., TOA or foreperiod), and applying a fixed duration, to see if this attenuates the AB. However, the current research seeks to investigate this by taking a temporal attribute of the task which is usually fixed (ITI) and varying it, to see if greater variability *increases* the AB by *reducing* temporal predictability, thus lessening readiness for the task.

## METHOD

### Participants

Thirty observers (21 female; age range 18–57, $M_{age} = 21.5$, $SD = 7.4$) provided written informed consent and participated voluntarily. Most were psychology undergraduates who were received course credit. Ethical aspects of the research were approved by the Australian National University Human Research Ethics Committee (protocol 2015/184), in accordance with the ethical standards of the 2008 Declaration of Helsinki.

### Design

A repeated-measures design was used. Participants completed both versions of the AB task in counterbalanced order in a single 30-minute session.

### Apparatus

Stimuli were presented on a 23.6 inch VIEWPixx liquid crystal display (LCD) with a refresh rate of 120 Hz and a resolution of 1,920 × 1,080 pixels. Behavioural responses for the AB were collected using a Cedrus RB-8 SUBJECT response box, with buttons labelled 2, 3, 4, 7, 8, and 9. Two buttons were left blank to enable participants to indicate if T2 did not appear or they did not perceive it. Experimental stimuli were created using Psychophysics Toolbox (version 3) for MATLAB R2012b (*Brainard, 1997*; *Pelli, 1997*; *Kleiner et al., 2007*).

### Stimuli

Stimuli were displayed in white (luminance 96.2 cd/m$^2$) on a grey (17.3 cd/m$^2$) background. Items were presented using an RSVP stream in 1.23° Helvetica font. Each trial commenced with a fixation cross and contained between 18 and 22 stimuli. Targets were numerals (2, 3, 4, 7, 8, 9) and distractors were uppercase letters, excluding those that could be confused with numbers (i.e., I, L, O, Q). Each stimulus in the stream appeared for exactly 100 ms (12 frames at 120 Hz). Most trials included two targets (T1 and T2), with some 'blank' trials including only one target. T1 appeared with a jitter of ±2 items, after 4-8 distractors, and T2 was presented at either lag 3 or lag 8. In both conditions, 30 trials were presented for each lag, with lags randomly interleaved. When each AB trial had concluded, participants were allocated 3.5 s in which to respond. If participants did not respond within the time frame, a non-response was recorded, and the next trial commenced. The ITI commenced immediately after the participant had keyed in their answer for T2, or after the time limit had elapsed.

During the *fixed-ITI* condition, the next trial always commenced 1500 ms after the participant's previous response had been made. During the *random-ITI* condition, ITI varied randomly between 100 and 3,100 ms after the previous response had been made. The random interval was set using MATLAB's internal rand.m function and PsychToolbox's "WaitSecs.m" (see code section of the Supplementary Materials).

### Procedure

Participants were given two AB practice blocks with auditory feedback. Both practice trials used a fixed ITI. A high-pitched beep indicated that the participant gave the correct response, while a low-pitched beep indicated that the participant gave an incorrect response.

The first practice block involved 12 trials at half speed (24 frames per second). The second practice block was at full speed. Following this, participants completed the two experimental blocks in counterbalanced order. Experimental trials took place in a darkened room to avoid peripheral distractions. To ensure consistency and minimise distraction, the experimenter was not present during the experimental trials. After completing both versions of the task, participants were asked verbally if they had detected any difference between the two versions of the task. Qualitative verbal responses were recorded. After answering this question, participants were verbally debriefed.

## Statistical analysis

Three dependent variables were analysed: T1 accuracy, T2|T1 accuracy, and blink magnitude. Of these, blink magnitude was the primary dependent variable of interest. T2|T1 accuracy represents accuracy at identifying T2 on trials where T1 was correctly identified, and was used to calculate blink magnitude. T2|T1 was compared between lags to confirm that an AB effect was observed. Blink magnitude was calculated by subtracting T2|T1 at lag 3 from T2|T1 at lag 8 ($T2|T1_{lag\ 8} - T2|T1_{lag\ 3}$), and was compared between fixed-ITI and random-ITI conditions. T1 accuracy was also compared between conditions, to ensure that the blink magnitude results could not be attributable to systematic differences in T1 identification. Statistical analyses were conducted in jamovi, version 9.5.15 (*The jamovi project, 2019*), and non-parametric tests were used where Shapiro–Wilk tests indicated assumptions of normality were violated. Verbal responses to the detection question were examined to determine whether the experimental manipulation had successfully evaded detection.

In addition, we included supplemental Bayesian analyses of the results, in order to give more information about null results. Unlike traditional NHST, Bayesian analyses can give information about the extent to which the data support the null hypothesis (*Quintana & Williams, 2018*). When evaluating support for the null in a Bayesian framework, it is more intuitive to use $BF_{01}$ (the simple inverse of $BF_{10}$), which in simple terms gives an odds ratio of how much more likely the null is than the alternative hypothesis (e.g., a value of $BF_{01}$ = 3.5 would indicate that the null is 3.5 times more likely than the alternative; *Quintana & Williams, 2018*). These analyses were carried out in JASP statistical software (*JASP team, 2019*), which provides a free, easy-to-use interface for these analyses. All data and analyses are available as supplementary materials to the paper.

# RESULTS

## T1 accuracy

Average T1 accuracy, expressed as proportion correct, was .88 (*SD* = .10, median = .89) for the fixed-ITI condition and .87 (*SD* = .11, median = .88) for the random-ITI condition. A Wilcoxon Signed Rank Test indicated there was no systematic difference in T1 accuracy between conditions, $W = 167$, $p = .829$, $M$ (diff) = .01, $SE$ (diff) = .03, 95% CI [−.04–.03], $\delta = .07$. In addition, we carried out a Bayesian paired-samples $t$-test analysis using JASP (*JASP team, 2019*), using a Cauchy prior of .707. This provided moderate evidence for the null, $BF_{01} = 4.79$, *error % = .009*.

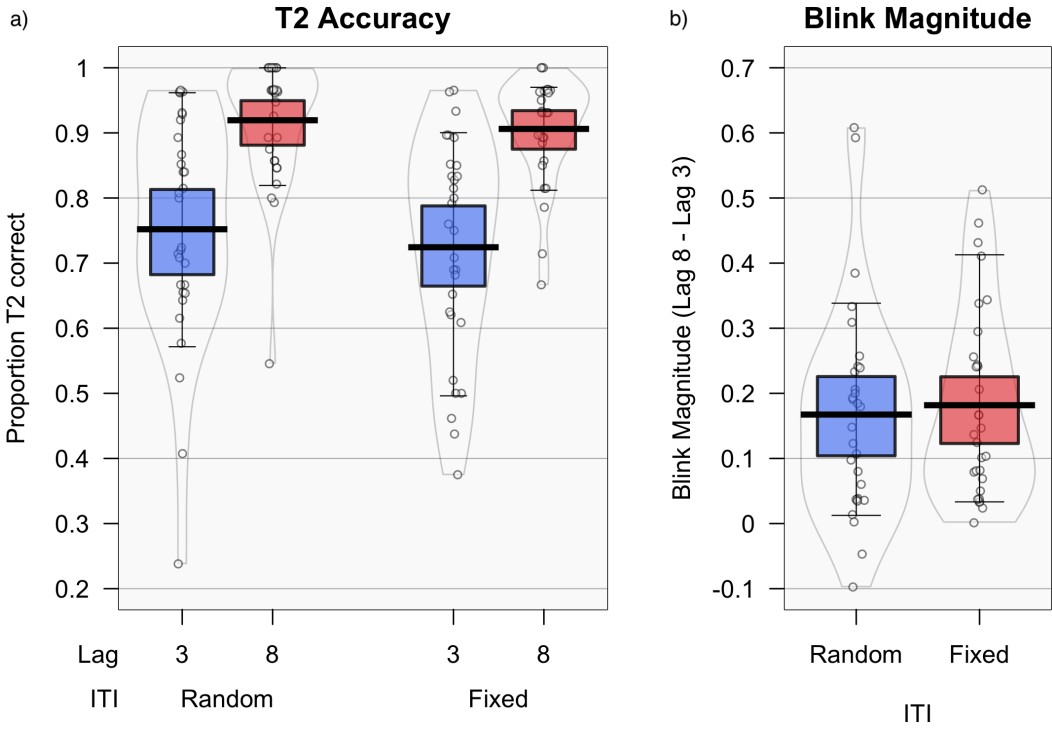

**Figure 3** **Accuracy and blink magnitude for random vs. fixed inter-trial intervals.** T2 | T1 accuracy (i.e., the proportion of T2 targets correctly identified for trials in which T1 was also correctly identified), comparing fixed and random ITI at each lag (A), and blink magnitude (B) (lag 8–lag 3). Means are shown by black horizontal bars. Individual scores are represented by black circles, slightly jittered for clarity; colored areas represent 95% Highest Density Intervals (HDIs), calculated using R's BEST (Bayesian Estimation Supersedes the T-Test) package (*Kruschke, 2013*), and vertical bars represent the 10th and 90th quantiles.

## Attentional blink

Figure 3 displays mean T2|T1 accuracy for lags 3 and 8 in each condition. Wilcoxon Signed Rank Tests confirmed there was a significant AB effect (i.e., identification of T2 was consistently lower for lag 3 compared with lag 8) in both the fixed-ITI condition, $W = 465$, $p < .001$, mean difference $= .17$, $SE$ (diff) $= .03$, 95% CI [.11–.23], $\delta = 1.02$, and the random-ITI condition, $W = 446$, $p < .001$, $M$ (diff) $= .15$, $SE$ (diff) $= .03$, 95% CI [.10–.21], $\delta = 1.02$.

A paired $t$-test (used since assumptions of normality were not violated in this case) indicated that blink magnitude did not significantly differ between fixed-ITI ($M = .18$ $SD = .14\%$) and random-ITI ($M = .16$, $SD = .16$) blocks, $t(29) = .49$, $p = .631$, $M$ (diff) $= .01$, $SE$ (diff) $= .03$, 95% CI [$-.04$–.07], $\delta = .09$. A follow-up Bayesian paired-samples $t$-test using JASP showed moderate evidence for the null, using a Cauchy prior of .707, $BF_{01} = 4.61$, $error \% = .007$.

## Order effects

As the trial order was randomised (either random ITI first or fixed ITI first), order effects were analysed in the results using a mixed ANOVA on blink magnitude, with ITI as the

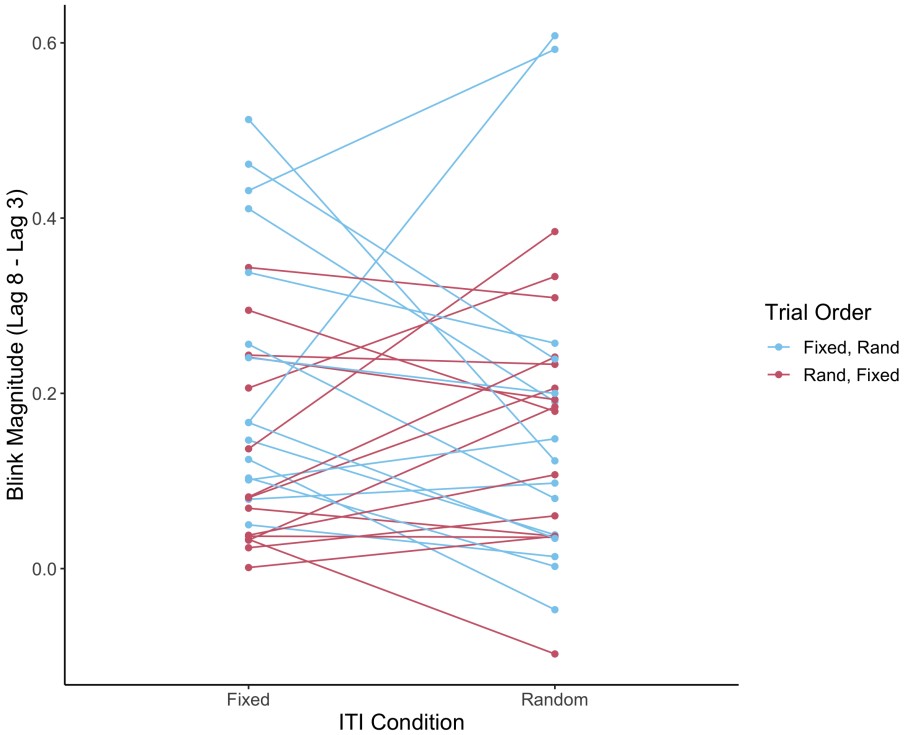

**Figure 4** **Individual subjects' performance on fixed and random ITI conditions according to trial order.** The plot shows blink magnitude (lag 8 accuracy–lag 3 accuracy), comparing fixed and random ITI for the different trial order conditions (fixed ITI first and random ITI first). Individual subjects are represented by the linked points; fixed-first and random-first conditions are indicated by color.

within-subjects factor and order as the between-subjects factor. There was no significant main effect of ITI (fixed vs. random), $F(1, 21) = 0.3$, $p = .617$, $\eta^2_p = .009$, or order, $F(1, 28) = 1.7$, $p = .202$, $\eta^2_p = .057$, and no significant interaction between trial order and ITI, $F(1,28) = 3.5$, $p = .072$, $\eta^2_p = .11$. However, it should be noted that this analysis had low power, as it was not initially included as part of our experimental design, so would be worth exploring in further research. The results are plotted in Fig. 4.

### Reaction times

Since reaction times are occasionally used as an indirect measure of task engagement (e.g., see *Brigadoi et al., 2018*), we also analysed the participants' reaction times for the first button press response in the task. Even though responses were not speeded, slower reaction times in the random-ITI condition might reflect increased task difficulty or reduced attention to the task. Reaction times were computed as median RT from the end of the RSVP stream to the first button press (response for T1) across both lags. A paired $t$-test (used since assumptions of normality were not violated) indicated that median reaction time (in seconds) did not significantly differ between fixed-ITI ($M = 1.001$ $SD = .199$) and random-ITI ($M = 1.009$, $SD = .189$) blocks, $t(29) = .30$, $p = .769$, $M$ (diff) $= .008$, $SE$ (diff) $= .026$, 95%CI $[-.045–.06]$, $\delta = .05$. A follow-up Bayesian paired-samples $t$-test

using JASP showed moderate evidence for the null, using a Cauchy prior of .707, $BF_{01} = 4.94$, *error* % = .011.

## Subjective reports

Qualitative verbal responses showed that 9 out of 30 participants did not detect any difference between fixed- and random-ITI blocks. An additional 19 participants identified incorrect differences (e.g., change in overall length, change in task stimuli). Only two participants correctly detected that one version of the task had randomised intervals between trials; one of these participants had extensive prior exposure of the original AB task through other research participation. This suggests that most participants were not consciously aware of the temporal difference between AB blocks.

## DISCUSSION

This finding does not show evidence that randomising the ITI affects performance in the AB task. A post-hoc Bayesian analysis shows moderate evidence for the null. This result expands upon findings made by *Badcock et al. (2013)* who found that randomising the ISI between trial commencement and T1 presentation did not affect AB performance when compared to an extended ISI (∼880 ms). The comparison of a random ITI with a fixed ITI (such as is more typically employed in a standard AB task), provides useful information regarding the attentional resources required to perform both tasks. The maintenance of accuracy during the random ITI condition suggests that these resources do not vary between conditions. In addition, the lack of differences in the reaction time data suggest that participants were most likely equally engaged during both types of task. Qualitatively, stable performance across the two trial types suggests that overall task difficulty, engagement, and motivation do not deviate when a randomised ITI is present.

Our analysis on order effects in the data was suggested by a reviewer, but since it was a post-hoc analysis we note that it is somewhat underpowered, so these results should be interpreted with caution. However, it is interesting to note that the learning effect from the first to the second session seems somewhat less in the random-first compared to the fixed-first group (see Fig. 4). This would be an interesting trend to follow up in future studies.

These findings have useful implications for future research, as it provides confirmation that the standard AB task, typically used as a measure of transient attention, has similar levels of interest and difficulty to those tasks which employ a random ITI. Such tasks are typically used as a measure of sustained attention (e.g., the Psychomotor Vigilance Test (PVT); *Dinges & Powell, 1985*) and the Continuous Performance Test (CPT); (*Cornblatt et al., 1988*). The current finding provides support for future research, which may seek to directly compare sustained and transient attention using the aforementioned tasks, without task difficulty/engagement confounding performance results.

The method adopted in the present experiment can be summarised as being a highly pared-back version of the AB task, in conjunction with a brief but randomised ITI, akin to the PVT, though considerably reduced in length. The decision to use the simplified version of the AB task was mainly practical; fewer lags required fewer trials overall, increasing

the number of trials possible for each lag and thus giving increased signal-to-noise ratio. However, the rationale for the use of the random interval was more complex.

Although most experimental paradigms involving a PVT use longer randomised intervals of 2 to 10 s, this was deemed too long in the current experiment, which instead used a randomised ITI of 0.1 to 3.1 s. Firstly, the inclusion of a longer ITI would drastically alter the length of the two versions of the task, potentially rendering them incomparable. Secondly, it was believed that the use of a longer ITI might appear too obvious in contrast with the fixed ITI, and that this noticeable difference might consciously influence participants' performance. Indeed, it appeared that the abridged ITI of 0.1 to 3.1 s did successfully evade detection in most cases.

It is possible that, during the randomised ITI condition, participants may have performed differently during longer compared to shorter ITIs –there is certainly precedent for this in the literature (e.g., see *Brigadoi et al., 2018*). In addition, performance on each trial might have been affected by the length of the previous trial's ITI (trial n-1); this might be expected from evidence in the foreperiod literature (e.g., see *Los, Knol & Boers, 2001*). Unfortunately, it was not possible to carry out these analyses with our current data, as we did not record the exact ITI for each trial. However, we have now updated the experimental code to include this variable, and it is available with the Supplementary Materials to facilitate replication and extension of this study.

Further studies could undertake to more clearly establish the perceptual limits associated with the incorporation of a randomised ITI. It is possible that, if imperceptible, a longer randomised ITI might more accurately represent a hybrid version of the PVT and AB tasks, and could therefore more accurately measure the possibility that task difficulty affects performance. Some research suggests much longer randomised ITIs (6,000 –10,000 ms) had a measurable effect on task performance and engagement in the SNARC effect (the spatial numerical association of response codes; *Brigadoi et al., 2018*), and indeed the random ITIs in sustained attention tasks do tend to be longer than 3,000 ms (our maximum ITI in the random condition). It would be useful to include these longer ITIs in future studies.

## CONCLUSION

The current findings may benefit future research which could seek to compare transient attention using the AB task, with measures of sustained attention that employ a random ITI, such as the PVT or CPT. Randomising the ITI on the AB task did not significantly impact performance, suggesting that both tasks may have similar levels of difficulty. This finding has broad implications for future research, expanding possibilities for comparing the effects of factors, such as level of arousal or affective state on these discrete forms of attention. This is valuable, given the wealth of existing research concerning both the AB and PVT (*Dux & Marois, 2009*; *Langner & Eickhoff, 2013*). Importantly, these findings support the legitimacy of comparing performance on the AB and random ITI tasks as a measure of transient and sustained attention respectively. This finding may be crucial in obtaining a more detailed understanding of attention, as it provides support for future research to

investigate the roles of sustained and transient attention in the same individuals, using these tasks.

### Funding
This study was supported by the National Health and Medical Research Council of Australia (grant no. APP1054726) to Deborah Apthorp. The funders had no role in study design, data collection and analysis, decision to publish, or preparation of the manuscript.

### Grant Disclosures
The following grant information was disclosed by the authors:
National Health and Medical Research Council of Australia: APP1054726.

### Competing Interests
The authors declare there are no competing interests.

### Author Contributions
- Lucienne Shenfield conceived and designed the experiments, performed the experiments, analyzed the data, authored or reviewed drafts of the paper, and approved the final draft.
- Vanessa Beanland conceived and designed the experiments, analyzed the data, authored or reviewed drafts of the paper, and approved the final draft.
- Deborah Apthorp conceived and designed the experiments, analyzed the data, prepared figures and/or tables, authored or reviewed drafts of the paper, and approved the final draft.

### Human Ethics
The following information was supplied relating to ethical approvals (i.e., approving body and any reference numbers):
The Australian National University Human Ethics Committee granted Ethical approval (2015/184).

### Data Availability
The raw data and the MATLAB code are available in the Supplemental Files.

### Supplemental Information
Supplemental information for this article can be found online at http://dx.doi.org/10.7717/peerj.8677#supplemental-information.

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
