# Peer review of "Temporal predictability does not impact attentional blink performance: effects of fixed vs. random inter-trial intervals"

_PeerJ, doi:10.7717/peerj.8677_

## Round 0.1 · original submission · Major Revisions

Your manuscript has been revised by three experts in the field. They expressed an overall positive view of the manuscript but indicated several points that you have to consider before the paper can be regarded as acceptable. Note that reviewer 3 has raised several concerns, including the statistical analysis, and also suggested some new analyses to be done. Reviewer 1 also did several critical questionings and is also available to discuss with you the suggestions/corrections needed.

·

Basic reporting

The article is clearly written and nicely presented. It’s great to see all of the individuals depicted in the violin plots. This gives a nice overview of the data.

Experimental design

The overview design of the experiment is clear but I think some more of the technical details should be included for replication purposes.
Line 153 – stimulus. Please include the timing of the stimulus presentation. I assume it’s not the 100 ms SOA described in Figure 1 as 273 ms is mentioned for lag 3 later on so maybe 91 ms? Important to know the details including fixation duration, any blank foreperiod etc. All of those details needed to replicate the methods.
Consider including these timings on Figure 1 rather than the standard 100 ms timings (for example). I think a reader could be forgiven for assuming that Figure 1 reflects what you did rather than a general description of typical tasks.

Validity of the findings

This is a simple and elegant design and the results and discussion clearly follow from this. There should be more research like it!

Additional comments

This paper reports on a simple but useful manipulation of inter-trial interval in a dual-target rapid serial visual presentation, estimating the attentional blink based on two inter-target intervals/lags: short (lag 3) and long (lag 8). Random or Fixed inter-trial intervals are manipulated in a counterbalanced, repeated-measures design. The analysis indicates equal performance between condition, supported by a Bayesian inferential test.
Dear Dr Apthorp and colleagues,
This is a great little project, well placed within the limited literature exploring this sort of preparation within the attentional blink literature, and nicely conducted with appropriate statistics and good use of Bayesian tests to examine null effects. Great to see JASP and Jamovi getting a run! And I appreciate that you talked to the participants about their experience – should be more of this.
I have a few comments that might have some bearing on the results and a few specific things that I think should be adjusted.
I hope you find the ideas helpful.
Best wishes,
Nic Badcock
I’ve arranged my comments in two parts: major comments related to larger suggestions for consideration and specific comments related to a specific line or figure.
Major comments
1. Order effects. Given that there’s counterbalancing in here, I think it would be valuable to test whether this impacts performance. Practice seems to make a difference after little exposure to these paradigms so I could imagine that the first conditions participants completed (a between-subjects test or mixed-ANOVA) would be most sensitive to the research question. Relatedly, I also note that the mean in Figure 3 for Lag 3 in the Random condition doesn’t touch any of the data and the underlying distribution is somewhat bimodal. I’ve recently seen this in some data I’ve been collaborating on in a completely different paradigm and when we broke it down by order, that bimodal distribution reflected the order of condition completion (lower scores for participants completing the condition first, higher for participants competing the condition second). Potentially order could be coded as different symbols in Figure 3 to get a quick idea of whether this matters but I think the interaction needs to be tested and reported. You might want to consider excluding the individual who’d had a lot of previous experience from this analysis.
Note: I just had opened the data as recommended as part of the review and noticed you have order in there – might actually be opposite to my suggestion, seems to be an advantage of doing the Random condition first. See what you think.
Also: There seems to be a duplicate data file with different names but the ‘peerj-42318-AB_TemporalPredictability_for_R.csv’ file has participant initials in there. I’d strongly recommend not including this for publication from an ethics-anonymity point of view but more generally, I’d only keep this identifiable information in a ‘key’ document that links a random id to the individual – best to avoid keep the data with any potentially identifying info.
2. Linear mixed effects. There’s a rich amount of variability from trial-to-trial in the Random condition. This could be examined in relation to T1 and T2|T1 accuracy in a linear mixed effects model, which would be a more sensitive test of the hypothesis: with the expectation that shorter inter-trials intervals would be associated with lower accuracy. I appreciate that this may be a novel analysis, often implemented using R, and so may represent some new learning but I think it could add significant value to the work so would be worth the investment.
3. N-1 effects. This is a final broad suggestion to make sure that we’re not missing anything. In the foreperiod literature (e.g., Los et al., 2001), there’s an influence of the foreperiod from the previous trial such that this seems to set expectations. If the n-1 foreperiod is long and the n foreperiod is short, performance (reaction time in particular) is poorer/slower, but if both are long, performance is good. In thinking about this, it might be nice to explore whether this has a role in the current data set using the above-suggested linear mixed effects approached by based on the timing of the previous trial. In fact, it’s probably worth running the n and n-1 timings within the same model as they could well interact as they do in the foreperiod literature.
Los, S. A., Knol, D. L., & Boers, R. M. (2001). The foreperiod effect revisited: Conditioning as a basis for nonspecific preparation1. Acta Psychologica, 106(1–2), 121–145. https://doi.org/10.1016/S0001-6918(00)00029-9
Specific comments
4. Line 152, there’s an extra reference for version 3 of the Psychtoolbox that might be nice to include:
5. Kleiner, M., Brainard, D., Pelli, D., Ingling, A., Murray, R., & Broussard, C. (2007). What’s new in psychtoolbox-3. Perception, 36(14), 1–16. Retrieved from https://nyuscholars.nyu.edu/en/publications/whats-new-in-psychtoolbox-3
6. Lines 200-202: Given that Bayes Factors are still potential unfamiliar to readers, it might be worth explaining this in more detail in the statistical analysis section of your methods. Including the criterion against which you’re judging the null to be supported would be helpful – I’m still used to seeing it in the other direction (i.e., <.03 is support for the null) but I understand that this is easily flipped. Probably more of a Bayes-advocacy role, helping people new to Bayes understand that the directions can change and sometimes > 3 can be considered support for the null.
7. Figure 2: Consider relabelling the y-axis to Percentage Correct. It’s just that the T2|T1 is a slight deviation from this. I’m not sure about this one – perhaps depend on the next comment
8. Figure 2: Relate to the above, I think it’s probably more accurate for the legend to reflect the task. This is the case for T2 Only but not for T2|T1 (which has two | by the way: “T2||T1” rather than “T2|T1”).
9. Figure 3 title: probably good form to add the citation for the BEST package

·

Basic reporting

This is a well-written paper which reports results of relevance in point of AB magnitude stability over ITI manipulation (fixed vs. variable). The paper is clear, concise, and to-the-point.

Experimental design

Just a question: how was the 100 -- 3100 ms jittered? In steps of -- like -- 100 ms? Or what? This should be specified in the method section. Related to this: given the fixed ITI was included in the variable-ITI range, one may wonder what could have happened by testing just the extremes of the temporal range in the variable ITI condition (100--500 ms and 2600--3100 ms, perhaps in steps of 100 ms...).

Validity of the findings

Although I am positively inclined towards acceptance of the present paper, I am wondering how the authors can conclude anything about subjects' engagement, task difficulty, and motivation (see General Discussion) without a hint of empirical evidence testing these various aspects. The AB effects obtained are quite small, and perhaps the task was not difficult enought to detect the possible effect of task difficulty. Secondly, I am not sure what the authors mean by engagement and less so by motivation.

To give an example of a different manipulation of ITI to explore its possible influence on the SNARC effect, the authors should consider:

Brigadoi, S., Basso Moro, S., Falchi, R., Cutini, S., & Dell'Acqua, R. (2018). On pacing trials while scanning brain hemodynamics: The case of the SNARC effect. Psychonomic Bulletin & Review, 25, 2267–2273.

in which my team and I, besides SNARC, tested several psychological aspects to show that particulay long ITI have in fact an effect on subjects' performance (using RTs).

Reviewer 3 ·

Basic reporting

Background and Introduction do give some context. However, the aim of the study remains opaque. Is it to investigate if an alternative approach to manipulating temporal predictability of T2 (and T1) affects AB performance similiar to those manipulations studied previously? Or is it to compare the influence of random ITI on the AB with its influence on tests of vigilance? Related, information on the rational of the ITI manipulation – crucial for the study – is rather restricted. I think that given the trial foreperiod findings, it can be predicted that manipulating ITI should have similar effects on the AB, and indeed, this has not yet been studied. The section Inter-Trial Interval (ITI) however only considers findings in vigilance tasks and leaves open why one would expect that the AB is affected by an ITI manipulation. Is it simply because AB, PVT and CPT all measure some aspect of attention or because the authors propose they all measure a particular aspect of attention that should be sensitive to an ITI manipulation? Is the key word here then maybe “temporal attention”, and if so, what is the concept of temporal attention in this context? Or does research suggest that AB and vigilance tasks are similarly affected by other variables, such as personality or pre-stimulus alpha, and therefore ITI should have the same effect? Details on how ITI affects reaction time and accuracy in PVT and CPT tasks are also missing. This information is however needed to put the AB results into context. Moreover, in how far it is relevant for the study that the AB task requires unspeeded responses (line 125) remains unclear.

The basic prediction is that the random ITI increases the AB. Given the aim of the study, what would be the implication of finding a difference between the two conditions? What would be the implications of not finding a difference?

The content of the first two sentences of the second paragraph of the ITI section (lines 119-125) is highly redundant.

Line 132: Additionally, … - In how far what follows is an additional aspect of the study remains unclear.

The last sentence in section The Current Study (line 134-135) is highly redundant to the third-to-last sentence (lines 130-132).

How were trials with non-responses to T2 treated in the analysis?

I was surprised to find participant’s initials in one of the provided data files.

Also, assuming that column Accuracy in supplemental table AB_TemporalPredictability contains T2IT1, I miss T1 accuracy data.

Typo line 269: may can

Experimental design

In how many steps was the ITI varied?

Validity of the findings

Discussion
“This finding suggests that randomizing the ITI of an AB task neither impairs nor improves performance” (Discussion, 228-229). The authors expected differences and therefore tested for it, they did not design the study to show that there is no effect. Therefore, all they can conclude is that they did not find evidence that randomizing the ITI affects AB task performance. Also, Bayesian t-tests indicate that evidence for the null is not particularly strong.

„This result expands upon findings made by Badcock et al. (2013) who found that randomising the ISI between trial commencement and T1 presentation did not affect
AB performance when compared to a brief or extended ISI (273ms or 880ms respectively).” (Discussion, 229-231). This contradicts the author’s summary of Badcock et al.’s findings in the Introduction: “Badcock et al. (2013) found that, compared with randomly variable foreperiods, having a predictable foreperiod attenuated the AB, but only for relatively long foreperiods (~880 ms).”

“This finding has useful implications for future research, as it provides confirmation that
the standard AB task, typically used as a measure of transient attention, has similar levels of interest and difficulty to those tasks which employ a random ITI [PVT and CPT]”. (Discussion, 238-240). I do not get this conclusion. How does not finding an effect of a random ITI for the AB confirm similarities with tasks in which – as summarized in the Introduction - an effect of a random ITI is normally seen? Moreover, what is meant by “similar levels of interest and difficulty”?

The authors should consider that even though on average accuracy does not differ significantly between conditions, the variable ITI might still affect performance. For instance, performance might be very poor for the short ITI conditions and much improved for the long ITI conditions, which would not show up in the averaged accuracies. This could be tested in an exploratory analysis.

I can follow the reasoning for using the abridged ITI in the present study. However, it should be discussed that this decision clearly restricts the comparability to the vigilance paradigms with much longer randomized ITIs. It also might be the reason for not finding a difference between the experimental conditions. The authors should also consider the possibility that their null finding indicates that within the AB, the onset of the RSVP stream is more relevant for the distribution of attentional resources and thus task performance than the ITI.

The final paragraph of the Discussion (160-266) needs more context. What is meant by “sensitivity thresholds around the current experimental paradigms”? How would a more accurate representation of a hybrid PVT and AB task help to measure the possibility that task difficulty effects performance? In turn, why should task difficulty not affect performance? What would be learned from measuring RT in a task that is unspeeded?

Conclusion
“Importantly, these findings support the legitimacy of comparing performance on the AB and random ITI tasks as a measure of transient and sustained attention respectively”. I cannot see how the authors arrive at this particular conclusion, and, more generally, the repeated claim that their results have broad implications for research comparing the AB as a measure of transient attention and other paradigms measuring sustained attention. Maybe this will become more clear if Introduction and Discussion are revised around a clearly described study aim.

Other
As previously mentioned, assuming that column Accuracy in table AB_TemporalPredictability contains T2IT1, I miss T1 accuracy data.

Additional comments

I agree with the authors that ITI is a factor in AB paradigms whose potential influence on task performance has been overlooked so far. The statistical analyses run and the description of results is comprehensible. The weakness of the manuscript is the lack of a clearly formulated objective, which is reflected in problems in the Introduction, Discussion and Conclusions outlined above.

---

## Round 0.2 · accepted · Accept

Thank you for your careful review of the original manuscript. I have looked at the text and I realize that you have made all the relevant changes required by the reviewers.